# Lot-to-Lot Variance in Immunoassays—Causes, Consequences, and Solutions

**DOI:** 10.3390/diagnostics13111835

**Published:** 2023-05-24

**Authors:** Yunyun Luo, Martin Pehrsson, Lasse Langholm, Morten Karsdal, Anne-Christine Bay-Jensen, Shu Sun

**Affiliations:** Biomarkers and Research, Nordic Bioscience, 2730 Herlev, Denmark; mpe@nordicbio.com (M.P.); lla@nordicbio.com (L.L.); mk@nordicbio.com (M.K.); acbj@nordicbio.com (A.-C.B.-J.); ssu@nordicbio.com (S.S.)

**Keywords:** immunoassay, lot-to-lot variance, quality control, critical quality attributes

## Abstract

Immunoassays, which have gained popularity in clinical practice and modern biomedical research, play an increasingly important role in quantifying various analytes in biological samples. Despite their high sensitivity and specificity, as well as their ability to analyze multiple samples in a single run, immunoassays are plagued by the problem of lot-to-lot variance (LTLV). LTLV negatively affects assay accuracy, precision, and specificity, leading to considerable uncertainty in reported results. Therefore, maintaining consistency in technical performance over time presents a challenge in reproducing immunoassays. In this article, we share our two-decade-long experience and delve into the reasons for and locations of LTLV, as well as explore methods to mitigate its effects. Our investigation identifies potential contributing factors, including quality fluctuation in critical raw materials and deviations in manufacturing processes. These findings offer valuable insights to developers and researchers working with immunoassays, emphasizing the importance of considering lot-to-lot variance in assay development and application.

## 1. Introduction

Immunoassays are extensively utilized in medical and scientific research for the detection and quantification of specific molecules, such as hormones, antibodies, and other proteins. These assays rely on the interaction between antibodies and their corresponding antigens, which can be measured using various labeling methods, including radioisotopes, gold nanoparticles, colorimetric, fluorometric, or chemiluminescent techniques. Despite their high sensitivity and specificity, immunoassays are not without limitations. One of the most significant challenges is the lot-to-lot variance (LTLV) in the assay results obtained from different batches or lots of kits [1].

LTLV, also known as batch variability, can arise due to several factors, including variations in the quality, stability, and manufacturing processes of key reagents, as well as storage and handling conditions of raw materials, etc [2,3,4]. LTLV has a significant impact on the accuracy, precision, and overall performance of immunoassays, which can result in inconsistent and inaccurate results [5]. This has serious consequences in the in vitro diagnostic (IVD) field, where accurate and reliable results are essential for patient care [6]. For instance, if an immunoassay with high LTLV is used to test a patient’s cardiac troponin I (cTnI) levels which is a gold standard biomarker for diagnosing myocardial infarction, it could lead to a wrong diagnosis or inappropriate treatment and ultimate result in a fatal outcome for the patients.

Based on our two decades of assay development experience, it is crucial to exercise careful control and monitoring of the quality of key reagents used in immunoassays to mitigate the impact of LTLV. This involves regular testing and verifying the purity, stability, aggregation, and activity of the reagents. Recently, Ario de Marco et al. put forward a straightforward set of guidelines, which, when correctly applied to protein and peptide reagents, can enhance experimental reproducibility. It should be noted, however, that these guidelines aim solely for the research field and not the IVD sector [7]. Adherence to rigorous quality control measures during the production processes can further aid in reducing the impact of this variability and guarantee consistent, precise, and dependable performance. 

## 2. Why LTLV Occurs?

At the technical level, the quality of immunoassays is determined by two key elements: raw materials and production processes. It is estimated that 70% of the immunoassay’s performance is attributed to the raw materials, while the rest 30% is ascribed to the production process (such as buffer recipes, reagents formulation, etc.).The production process guarantees the lower limit of kit quality and reproducibility, while raw materials provide the foundation for the sensitivity and specificity of IVD kits, thereby forming the upper limit of kit quality, akin to the icing on the cake. To pinpoint the causes and origins of LTLV, it is necessary to explore both of these aspects in detail. Additionally, the instability of the analyte epitope, to some degree, also contributes to LTLV.

### 2.1. Quality Fluctuation of Raw Materials

An immunoassay typically comprises various components, such as solid phases (e.g., magnetic particles, microtiter plates), antibodies, antibody conjugates, antigens, antigen conjugates, calibrators, kit controls, and assay buffers or diluents. Therefore, LTLV could potentially arise from any of these aforementioned constituents. Previous experience has shown that fluctuations and instability in immunoassays are largely related to the quality of raw materials, which are inherently biologics that are difficult to regulate, such as antibodies sourced from hybridoma. Moreover, other external materials, such as the master calibrator and lot-to-lot QC panel, may also impact LTLV (Table 1).

#### 2.1.1. Antigens

Antigens are molecules that trigger the immune system to generate antibodies against them, and they can be comprised of various substances such as proteins, polysaccharides, lipids, chemicals, or nucleic acids. In the context of immunoassays, the quality of antigen raw materials is critical, but a uniform standard for performance evaluation is currently lacking. Generally, antigen activity, purity, batch-to-batch consistency, and stability are the key evaluation criteria. Antigens are typically provided as clear and homogeneous liquids or lyophilized white powders that are free of contaminants, turbidity, and particulates. Sodium dodecyl sulfate–polyacrylamide gel electrophoresis (SDS-PAGE) is a standard method for assessing antigen purity and molecular weight, often followed by staining with Coomassie brilliant blue or silver. Additionally, a size exclusion column combined with high-performance liquid chromatography (SEC-HPLC) can also be used to determine purity and molecular weight. If the purity of the antigen is compromised, the efficiency of labeling may be impacted, leading to reduced specificity, signal, and increased background. Some antigens are unstable and require specific storage buffers that include protein stabilizers, such as bovine serum albumin (BSA) [8], urea, and glycerol [9]. Most antigens can be stored at −20 °C or −70 °C, and they should be frozen after being diluted and aliquoted. For synthetic peptides used as calibrators or quality controls, it is crucial to note that different batches of peptides may have varying amounts of the target peptide content due to truncated by-products from the synthesis process, although the gross peptide content may still be the same. 

#### 2.1.2. Enzymes

Horseradish peroxidases (HRP) and alkaline phosphatase (ALP) are enzymes that are commonly employed in IVD reagents. Enzymes are one of the few substances that can be measured using “activity units” rather than purity and mass. Horseradish contains at least seven isozymes of HRP, with isozyme C exhibiting the highest catalytic activity, accounting for up to 50% of the peroxidase content of horseradish [10]. Thus, it is not always necessary to utilize the most purified form of HRP, as this may lead to elevated background noise in certain assays. HRP and ALP are typically obtained via extraction from native materials, such as horseradish root and calf small intestine, respectively, and are then subjected to complex purification processes to obtain the required enzymes [10]. Given the reasonable cost limitations, a purity of 90–95% is typically deemed acceptable with current purification techniques. However, it is difficult to ascertain the number of biologically active enzymes with correct structures within this 5% impurity, and the presence of any interfering or inhibitory ingredients is sometimes unknown. As such, accurately controlling enzyme quality can be challenging. Although enzyme purity is consistent across batches, there are often notable differences in enzymatic activity. Thus, care must be taken when selecting enzyme manufacturers who claim to have improved production processes and have passed quality inspections, as such modifications may also affect other biological activities of the enzyme, ultimately impacting the expected assay performance.

#### 2.1.3. Antibodies

Monoclonal antibodies of high quality are crucial for the production of reliable reagents. However, modifications and labeling processes may affect the performance of the final product. Unfortunately, there is no standardized method to measure antibody quality, although activity, concentration, affinity, homogeneity, specificity, purity, and stability are commonly evaluated. Aggregation of high-concentration antibodies, particularly IgG3 [11,12,13], is a major issue that can be detected and separated using SEC-HPLC. Aggregates, fragments, and unpaired chains of antibodies can lead to high background and signal leap, causing overestimated analyte concentrations in the sandwich (Figure 1A) and indirect immunoassays (Figure 1C) and underestimated levels in competitive immunoassays (Figure 1B). Antibody labeling efficiency is related to purity, and impurity proteins can negatively impact the assay’s specificity, signal, and background. Antibody purity analysis, using methods such as SDS-PAGE, SEC-HPLC, and capillary electrophoresis sodium dodecyl sulfate gel electrophoresis (CE-SDS), is critical for successful immunoassay development. The cell culture process and antibody purification process largely determine antibody purity, and the use of a serum-free medium can remove impurities brought by fetal bovine serum. 

In recent experiments conducted in our laboratory, we observed a significant disparity when exchanging a monoclonal antibody directed towards CTX-III [14] from hybridoma to recombinant antibody, despite their identical amino acid sequences. We found that the recombinant antibody led to substantially lower sensitivity and maximal signals in the CTX-III assay when compared to the hybridoma antibody (as illustrated in Table 2). The recombinant antibody’s SEC-HPLC purity was observed to be approximately 98.7%, which is considered adequate (as depicted in Figure 2a). Nonetheless, CE-SDS analysis of the recombinant antibody exposed nearly 13% impurity (as shown in Figure 2b), causing a reduction in sensitivity and maximal signal. The primary impurities detected in recombinant IgG include a single light chain (LC), a combination of two heavy chains and one light chain (2H1L), two heavy chains (2H), and nonglycosylated IgG. 

#### 2.1.4. Antibody and Antigen Conjugates (Enzyme, Biotin, Fluor, etc.)

Some commercial protein conjugation kits can rapidly produce conjugates within an hour, which is advantageous in terms of saving time and resources. However, despite their seemingly straightforward process, “mix and use” conjugation products are known to be problematic due to the inefficiency of conjugation chemistry and the absence of a purification step. As a consequence, unreacted biomolecules and excess labels, such as fluorophores, biotins, and enzymes, may remain in the reaction mixture. In our internal evaluation, we observed that approximately 30% of monoclonal antibodies were unlabeled using a rapid commercial HPR conjugation kit (Figure 3a), whereas the same antibodies were fully labeled but resulted in excessive free HRP in another batch of conjugation (Figure 3b). Conversely, all antibodies were conjugated and likely with the same incorporation ratio when we switched to a standard HRP labeling kit (sodium periodate oxidation method) (Figure 3c).

#### 2.1.5. Kit Controls and Calibrators

Kit controls are typically produced using the same process and constituents as the kit calibrators, leading to a lack of independence that renders the kit control unsuitable for monitoring the integrity of the calibrator. In the event of instability arising from the control/calibrator materials or process, the kit control will shift to the same degree as the kit calibrator due to their identical composition, potentially giving a false sense of security regarding test performance. Any degradation of these materials would occur in both the controls and the calibrators, making it appear as though the test was still functional while it had actually shifted. In essence, kit controls serve as an additional set of calibrators or “pseudo controls”. For instance, our laboratory observed that certain laboratory-developed tests (LDTs) had acceptable shelf-life in the kit controls and calibrators, which were created using the same synthetic peptides, whereas the serum QC-panel failed to meet the criteria (data not yet published).

The use of pooled patient serum as kit controls by many manufacturers is a widely employed practice, but it raises certain overlooked issues, particularly concerning the consistency of manufacture. The lack of a standardized procedure for producing patient pool controls leads to significant variations in the manufacturing process, which, in turn, affects the stability and consistency of the product from one vial to another. The validation of the manufacturing process, dispensing, stability, and vial-to-vial homogeneity, among other factors, is typically not carried out for pooled patient biofluid controls.

#### 2.1.6. Buffer/Ultrapure Water

The assay buffer plays a crucial role in immunoassays as it serves as a diluent for various components, including calibrators, samples, antibodies, and antigens. It contains a mixture of proteins, salts, detergents, coloring agents, preservatives, and ultrapure or de-ionized water, which is carefully formulated to block non-specific interactions and maintain a stable pH environment for antibody and antigen immunoreaction. The optimization of assay buffers aims to minimize background signals and achieve high assay sensitivity. The conductivity of the water used in kit production can vary (e.g., 0.05–0.8 μS/cm) across different locations and impact the performance of the immunoassay. 

#### 2.1.7. Others (Containers, Microtiter Plates, etc.)

In the early stages of assay development, it is important to pay attention to small consumables that may pose unexpected challenges during product implementation. The quality and cleanliness of tubes containing reagents must be assessed thoroughly to ensure their usability and compliance with cleanliness standards. The presence of residual pollutants in tubes may vary across different products, underscoring the need to verify the qualification of tube materials through experimental design. A limited number of sampling inspections are insufficient to meet evaluation requirements, and more reliable results can be obtained by conducting multiple batch and sampling tests.

### 2.2. Deviation in the Production Process

The quality of IVD products is largely influenced by the production process of immunoassays. The 4P1E (people, products, procedures, premises, and equipment) or 4M1E (men, materials, methods, machines, and environment) are the five crucial elements that significantly impact the quality of the final products. Each element encompasses various factors, such as the skills and experience of the staff, the quality of raw materials, the precision of the production process, the suitability of the production facility, and the efficiency of the production equipment. Thorough attention to these factors is crucial for ensuring the reproducibility, reliability, and accuracy of immunoassays.

#### 2.2.1. People/Men

In the production of immunoassays, the quality of reagents may be compromised due to various factors during production processes, such as buffer preparation, conjugation, coating, blocking, drying, lyophilization, etc. Even if the same operator performs these tasks at different time points, it is challenging to ensure the reproducibility of the produced reagents. The introduction of personnel changes and the variability of operation tasks further exacerbate the uncertainty associated with the production process, leading to potential compromises in the quality of the reagents. 

#### 2.2.2. Products/Materials

The task of controlling incoming materials can prove challenging, especially when introducing new batches of antigens and antibodies or modifying key material components, even from different suppliers. As such, a comprehensive verification process must be conducted akin to that of producing new products. Even when replacing microtiter plates, magnetic beads, nitrocellulose membranes, and other consumables, resulting product quality may vary. Notably, in 2016, Life Technologies A/S discontinued the use of diglyme in the manufacturing of Dynabeads due to its classification as a reproductive toxin and listing in Annex XIV of the Registration, Evaluation, Authorization and Restriction of Chemicals (REACH) Regulation by the European Chemical Agency [15]. Consequently, the European Union market cannot use diglyme, and customers such as Roche Diagnostics GmbH had to assess alternative solvents coated on new magnetic beads [16]. However, some IVD assays yielded unsatisfactory reproducibility results, leading Roche to apply for authorization for the future use of diglyme in the EU [17].

#### 2.2.3. Procedures/Methods

Buffer preparation serves as a pertinent example of general preparation procedures, and although it is commonplace in most laboratories, it requires careful execution to avoid errors and is often a time-consuming and labor-intensive process. Buffer preparation entails several steps, including weighing the components, dissolving and mixing the reagents in a suitable container, adjusting and checking pH, and adjusting the final volume. During the mixing process, substance parameters such as density and viscosity bear great significance as they determine the energy requirements, miscibility, dispersing degree, and mixture stability. However, the complexity of buffer preparation has raised several questions that must be addressed, such as the optimal mixing duration and which device to use for different volumes. In general, for a one-liter solution, mixing should last for at least an hour, with the choice of the device depending on the buffer volume, where a magnetic stirrer is suitable for less than five liters and an overhead stirrer for volumes exceeding ten liters. Our lab recently noticed that a sandwich assay showed nearly 40% higher background when reproducing a new batch of eight litters of assay buffer, resulting in some samples falling below the lower limit of quantitation (LLOQ). This is found to be caused by the insufficient dissolving of chemicals in the buffer (unpublished data). Additionally, the order of adding reagents must be followed strictly, starting with inorganic salts, followed by detergent (e.g., Tween-20) and preservatives, and finally, inert proteins such as BSA. Dissolving Tris-base first and lowering its pH to the target before adding BSA is crucial when making Tris-HCl buffer to prevent precipitation, as high pH (e.g., pH 10–11) Tris solutions may cause BSA to precipitate.

#### 2.2.4. Equipment and Premises/Machines and Environment

The correlation between weather conditions and the performance of certain equipment, such as dehumidifiers, has been identified as a challenging factor in ensuring consistent drying quality. In particular, during rainy, foggy, cloudy, and sunny days, the same level of drying quality cannot be guaranteed due to the impact of these weather conditions. For strip test manufacturers, dispensing antibodies on nitrocellulose membranes during high humidity (>60%) can lead to wider T and C lines, resulting in smear issues. Conversely, low humidity (<40%) can cause uneven scribing and satellite points on the membrane due to strong electrostatic interaction. To overcome these issues, a recommended best practice is to balance the equipment and nitrocellulose membranes for several hours in an environment with a humidity level of 50% before dispensing. This approach is deemed necessary to ensure consistent and reliable performance of the equipment and membranes, regardless of the prevailing weather conditions.

### 2.3. Unstable Analyte Epitope

#### 2.3.1. Proteinase Cleavage of the Analyte

The structural and stability properties of numerous analytes remain largely unexplored, leading to potential degradation by proteinases present in blood circulation. Peptidyl Peptidase IV (DPP IV), Neprilysin (NEP), Corin, and the Insulin-degrading Enzyme (IDE) have been identified as some of the proteinases capable of cleaving specific amino bonds, including Pro-Lys, Met-Val, Gly-Cys, Arg-Ile, Lys-Met, Leu-Arg, and Arg-Arg [18]. Such cleavage events generate issues with LTLV in immunoassays as the analyte level in QC panels may gradually decrease over time. The proteolytic degradation of brain natriuretic peptide (BNP) into various fragments by enzymes such as DPP IV [19], NEP [20,21], and IDE [22], as well as the hydrolysis of cTnI [23,24] at the N- and C-terminals, exemplify the impact of these proteinases on analyte stability. This proteolytic degradation is not limited to in vivo conditions, as it can also occur in vitro in blood samples. The existence of various truncated forms of BNP and cTnI poses significant challenges to the accuracy of detection, highlighting the need for further research to address LTLV issues in immunoassays.

#### 2.3.2. Glycosylation

Glycosylation modifications are estimated to occur in nearly 50% of proteins and have been identified as a means of enhancing analyte stability against cleavage. However, the presence of glycosylation on the epitope of interest may also interfere with antibody and antigen interactions, leading to an underestimation of analyte concentrations [25]. The lack of international standards for some glycosylated analytes has resulted in the use of homemade master calibrators in many commercial immunoassays. Typically, these calibrators are nonglycosylated, but most antibodies used in such assays are directed against potential glycosylation sites on antigens that are nonglycosylated. This discrepancy can lead to a commutability problem between calibrators and clinical samples [26]. The heterogeneous glycosylation patterns observed on some analytes from different human specimens [27,28] may also contribute to LTLV. Therefore, the impact of glycosylation on the performance of immunoassays requires further investigation to establish standardized protocols for calibrator preparation and antibody production that account for potential glycosylation sites. Such efforts will ultimately improve the accuracy and reliability of glycosylated analyte measurements in clinical settings.

## 3. How to Minimize LTLV?

### 3.1. Antigen Design

Antigen design is an essential aspect of developing reliable immunoassays for clinical applications. To mitigate potential issues, epitopes with known glycosylation sites should be avoided. However, if a glycosylated epitope is clinically significant, it is feasible to synthesize or express a glycosylated antigen for antibody development. Amino acids such as Pro-Lys, Met-Val, Gly-Cys, Arg-Ile, Leu-Arg, Arg-Arg, or Lys-Met should be avoided within the epitope, as they can be cleaved by various proteinases in the blood. There is an exception where glycosylation residues in the epitope can prevent proteinase processing by blocking the enzyme’s access to the glycosylated cleavage site [25]. Moreover, caution should be exercised when selecting antigenic epitopes where the aforementioned amino acids are 10–30 amino acids apart. This is because potential cleavage may eliminate the spatial obstruction, leading to an increase in analyte levels in clinical samples over time, thereby facilitating antibody and antigen interaction. 

### 3.2. Critical Quality Attributes of Antigen and Antibody

The optimization of immunoassay LTLV through the production process is limited by current diagnostic technology, thus highlighting the significance of raw materials. Antigens and antibodies are the fundamental components that dictate immunoassay performance, each with its unique specificity, affinity, surface charge, and solubility properties. To ensure consistency in antigen and antibody batches, a comprehensive quality control analysis is essential, including monitoring concentration, immunoreactivity, charge isoform profile, purity, homogeneity, and more. Critical quality attributes that must remain consistent from batch to batch include isoelectric point (pI), charge profile, Ig class and subclass, stability, and kinetic constant. Failure to control these attributes can result in various consequences such as re-optimization of antigen/antibody amounts for each new IVD kit lot, sensitivity issues, and the need to re-optimize the labeling/coating process pH or volume, and removal of aggregates, fragments, or impurities.

#### 3.2.1. Minimize Antigen and Antibody Aggregation

To minimize the aggregation of proteins, it is recommended to keep the stock antibody at a relatively low concentration, as high protein concentrations are prone to aggregation. However, low concentrations can lead to adsorption and loss on the containers. In some cases, a higher protein concentration is necessary for conjugation purposes, such as HRP labeling. Therefore, it is ideal to add stabilizers to the storage buffer to minimize aggregation. Two major types of stabilizers can be added to the buffer based on their properties: kosmotropes and chaotropes. Kosmotropes, such as sucrose, glycerol, glycine, arginine [9,29], and sorbitol [30], stabilize the native structure of the protein by stabilizing the hydration shell around the protein(Figure 4a). Chaotropes, such as guanidine hydrochloride and urea [9], reduce or hinder protein-protein interactions by interfering with the hydration shell around proteins and weakening it (Figure 4b) [31]. Larger additives such as PEG or nonionic detergents can directly shield hydrophobic sites on proteins, thereby preventing their interaction with hydrophobic sites on neighboring proteins [32]. Other commonly used additives include arginine and citrate. Table 3 provides a summary of popular additives and their suggested concentration intervals for minimizing protein aggregation. As different additives act through different mechanisms, it is advisable to try several additives to determine which one yields the best result. SEC-HPLC can be used to remove a high number of protein aggregates.

#### 3.2.2. Regularly Monitor the pI of Antigen and Antibody

The heterogeneity of protein charge profiles can lead to the formation of protein aggregates and subsequent loss of sensitivity in immunoassays. This phenomenon is particularly evident at the pI, where the proteins carry no net charge and thus repel each other, resulting in precipitation. Capillary isoelectric focusing (cIEF) is an analytical technique used to generate charge isoform profiles, which are unique fingerprints of antigens and antibodies. Characterization of the pI and charge isoform profiles of these critical components is essential for proper buffer selection during assay development [33], as well as for maintaining the stability and consistency of the reagents. To this end, it is recommended to determine the pI range for each antigen and antibody across at least ten batches, with acceptance criteria including the similarity of the IEF profile of a new batch to a reference profile and alignment within the specific pI range determined from the ten lots.

#### 3.2.3. Accurate Quantification of Antigen and Antibody

Various methods are available for protein quantification, such as UV spectrophotometry (UV 280), the Folin-phenol method (Lowry method), Bicinchoninic acid assay (BCA), and the Bradford method [34]. These techniques have distinct principles and possess their own strengths and limitations. UV280 is commonly used for protein quantification by calculating the molar extinction coefficient based on the amino acid sequence [35], but it is prone to interference from UV-absorbing impurities such as trace nucleic acids, leading to inaccurate protein concentrations [36]. To ensure accurate protein quantification, it is recommended to use multiple methods, such as UV280 and BCA, for each batch of antigen, enzyme, antibody, and conjugate. Additionally, it is necessary to establish individual quantitative calibrators for each protein, which can prevent errors in protein quantification caused by any single method and ensure consistency between different batches of the same product. This approach results in minimal batch variation and consistent formulation in the final immunoassays.

#### 3.2.4. Purity Determination of Antigen and Antibody

CE-SDS is a recommended method to determine the purity of antigens and antibodies, whereas specific activity is a useful metric to assess the relative purity of enzymes. Additionally, Mass Spectrometry (MS) and HPLC can assist in detecting the presence of contaminating proteins, sample proteolysis, and minor truncations [34]. The identity of a sample can be confirmed using two different methods: ‘top-down’ MS, which examines the intact protein mass to identify any degradation during purification, and ‘bottom-up’ MS, which employs mass fingerprinting or tryptic digests to confirm the correct protein is being used, thus avoiding potential errors, such as the use of a host protein with similar mass that was accidentally purified [7].

#### 3.2.5. Homogeneity Determination of Antigen and Antibody

To ensure the quality of reagents, it is crucial to assess their homogeneity, which refers to the size distribution of antigens and antibodies and is closely related to the presence of aggregates. While polydispersity does not necessarily indicate instability, the detection of higher aggregates in the preparations suggests that the protein is not in an optimally functional state. This can lead to erroneous experimental results, such as an overestimation of the concentration of active protein during antigen-antibody interaction studies [7]. To detect homogeneity, several analytical techniques, including SEC-HPLC, size exclusion chromatography coupled with multi-angle light scattering (SEC-MALS), and Dynamic Light Scattering (DLS), can be employed [34]. Among these, SEC-HPLC is the most commonly used system for characterizing protein samples, as it can effectively separate and detect aggregates, fragments, and unpaired chains of antibodies. By determining the relative abundance of monomers versus other forms, this technique can provide valuable information on the composition of protein samples. A high degree of homogeneity indicates good stability of the reagent, and it is recommended that qualified antigens and antibodies contain at least 95% monomers.

#### 3.2.6. Maximize the Antigen and Antibody Storage Stability

In protein science, stability refers to the ability of a protein to maintain its functionality for a specified duration of time under particular storage conditions. The stability of a protein directly correlates with batch consistency and reduction in variability. The standard solvent used for storing most antigens and antibodies is the neutral PBS buffer. However, given the different properties and pI of proteins, they may undergo degradation through various pathways, including polymerization, denaturation, inactivation, and interaction with packaging materials [37]. Therefore, it is necessary to establish a unique storage buffer for each antigen and antibody to ensure optimal stability. To this end, our laboratory evaluated the stress stability of two hybridoma antibodies (PRO-C3 [38], PRO-C6 [39]) and one recombinant antibody (CTX-III [14]) in eleven different buffers. Notably, conventional PBS buffer at pH 7.4 did not provide the best stability for any of the proteins tested after they were subjected to stress at 37 °C for 14 days (Figure 5). 

#### 3.2.7. Oriented Conjugation on Antigen and Antibody

Antigens and antibodies are frequently conjugated with various molecules at random sites and orientations, which may affect their immunoreactivity and introduce considerable lot-to-lot variability (LTLV) in the final in vitro diagnostic (IVD) kit. However, the conjugation of small molecule drugs to specific glycosylated sites on antibodies has become increasingly popular for generating relatively homogeneous preparations of antibody-drug conjugates with consistent physicochemical properties, such as labeling ratio and sites across batches [40,41]. These oriented conjugation methods could also be employed for antigens and antibodies to reduce the LTLV of critical reagents [42,43]. 

### 3.3. Kit Control and QC-Panel

To reduce the LTLV in immunoassays, various measures, such as adding proteinase inhibitors and preservatives to the QC panel or freeze-drying the entire panel, can be employed. Aprotinin [25], benzamidine, and 4-benzene sulfonyl fluoride hydrochloride (AEBSF) [44] have been reported to be effective protease inhibitors for protecting BNP from degradation during sample storage. Custom controls, which are manufactured with materials and processes different from those of kit calibrators, can monitor assay integrity and LTLV effectively. Third-party controls offer the advantage of being independently manufactured from the assay reagents, providing additional confidence in the control. The control should be sensitive enough to detect shifts caused by reagent or kit lot changes, allowing corrective actions to be taken before reporting erroneous patient values.

### 3.4. Consistent Preparation Parameters for Buffer and Solution 

To ensure the reproducibility of immunoassays, it is essential to maintain consistency in stirring speed, time, and temperature during different batches. As the viscosity and pH of some buffers vary at different temperatures, it is recommended to use the same conditions for all batches. In-process testing of physicochemical parameters such as temperature, conductivity, OD280, pH, and osmolality (for serum-based kit controls and calibrators) should be conducted to ensure thorough mixing of the buffer. Overmixing should be avoided to prevent mechanical shear damage to the antigen and antibody solution. To establish acceptable specifications, it is suggested to record the physicochemical parameters of at least three batches and determine the means as the reference.

## 4. Discussion

The variability of immunoassays is influenced by multiple factors, including the quality and consistency of the reagents or kits utilized, which can have significant implications for assay performance if not properly formulated or handled. The storage and handling of reagents can also affect assay stability and reliability. Furthermore, the assay protocol itself is a significant contributor to LTLV, particularly when different laboratories or operators use different techniques and instrumentation. Consistent adherence to the assay protocol is therefore critical for minimizing variation and ensuring reliable and accurate results. 

The maintenance of accurate laboratory data over an extended period is essential for the appropriate treatment of patients and effective disease management. The standardization of measurements and implementation of a traceability system are necessary to ensure accurate results. Reference materials (RMs) are the fundamental components of these reference systems and play a critical role in establishing the traceability of the reference system [45]. RMs are used to calibrate, validate, and quality control the method and establish metrological tracing to compare inter-laboratory measurements. Moreover, it enables the linking of individual data from the laboratory through a continuous calibration chain to international measurements. However, not all commercial immunoassays have available RMs, creating challenges when bridging a new batch of immunoassays to a previous batch. Therefore, detailed characterization of an analyte’s structure and metabolism is essential to establish standard RMs and understand the potential reasons for LTLV in measurements.

To mitigate the problem of LTLV in immunoassays, researchers should utilize validated assay protocols, such as CLSI EP26 [46,47], and carefully evaluate the quality of reagents and kits [2]. The inclusion of appropriate controls and replicate measurements is also recommended to ensure the accuracy and reliability of results [48]. Researchers should be aware of the potential impact of LTLV on their experiments and take steps to minimize its effects, such as using multiple lots of reagents [49] or considering reagents from multiple manufacturers [5]. Additionally, the use of competitive assays for small molecules may lead to lower specificity and increased LTLV compared to sandwich assays. However, advancements in phage display technology have made it possible to develop sandwich assays against small molecules, as seen with Fujirebio’s sandwich 25-OH Vitamin D chemiluminescent immunoassay [50,51,52]. In conclusion, addressing LTLV is crucial to improve the accuracy and reliability of immunoassays, leading to more dependable biomedical research and diagnostic results. 

## Figures and Tables

**Figure 1 diagnostics-13-01835-f001:**
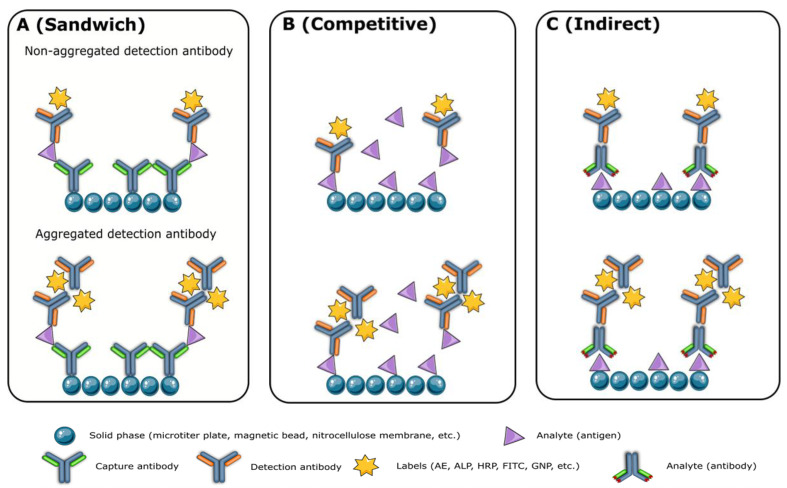
The impact of antibody aggregates on the sandwich (**A**), competitive (**B**), and indirect immunoassays (**C**). AE: acridinium ester; ALP: alkaline phosphatase; FITC: fluorescein-5-isothiocyanate; GNP: gold nanoparticles; HRP: horse radish peroxidase.

**Figure 2 diagnostics-13-01835-f002:**
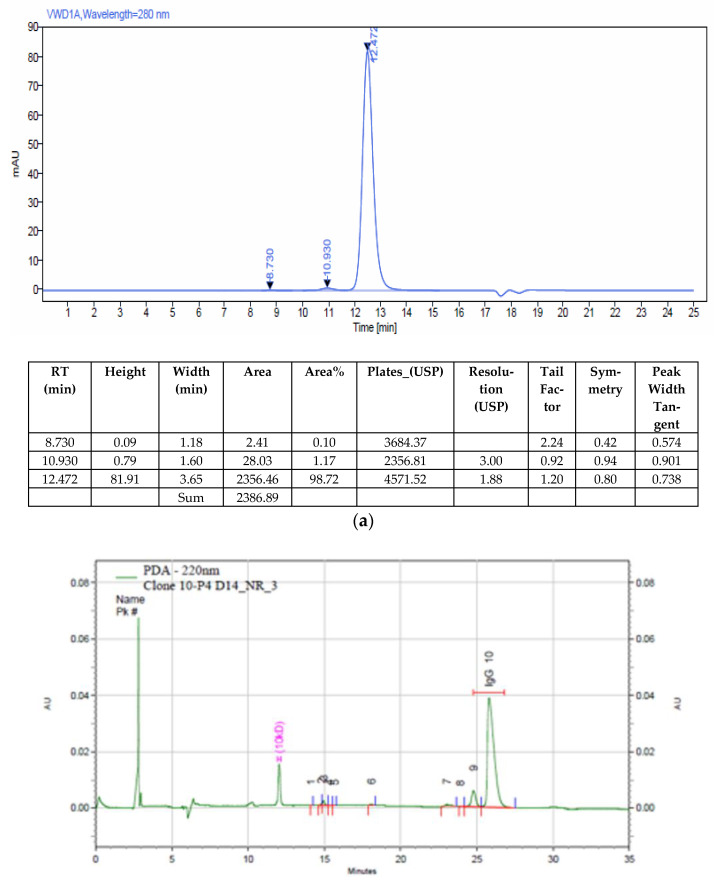
The purity determination of a recombinant monoclonal antibody against CTX-III by SEC-HPLC (**a**) and CE-SDS (**b**). CTX-III: C-terminal cross-linked telopeptides of type III collagen; SEC-HPLC: size exclusion column combined with high-performance liquid chromatography; CE-SDS: capillary electrophoresis sodium dodecyl sulfate.

**Figure 3 diagnostics-13-01835-f003:**
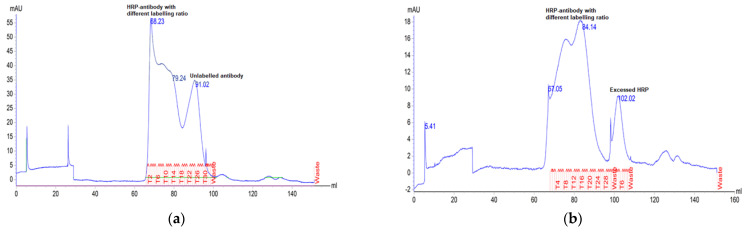
The purity determination of HRP-labeled monoclonal antibodies by SEC−HPLC. (**a**) antibody was labeled with a rapid HRP labeling kit (batch 1); (**b**) antibody was labeled with a rapid HRP labeling kit (batch 2); (**c**) antibody was labeled with a regular HRP labeling kit. HRP: horse radish peroxidase; SEC−HPLC: size exclusion column combined with high-performance liquid chromatography.

**Figure 4 diagnostics-13-01835-f004:**
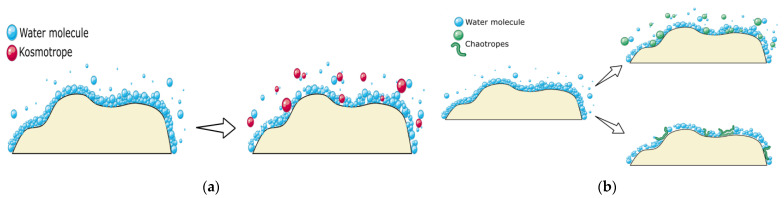
Two major additives to minimize the aggregation of protein are kosmotropes (**a**) and chaotropes (**b**). Adapted from Cytiva online course: Tips and troubleshooting: recombinant protein production.

**Figure 5 diagnostics-13-01835-f005:**
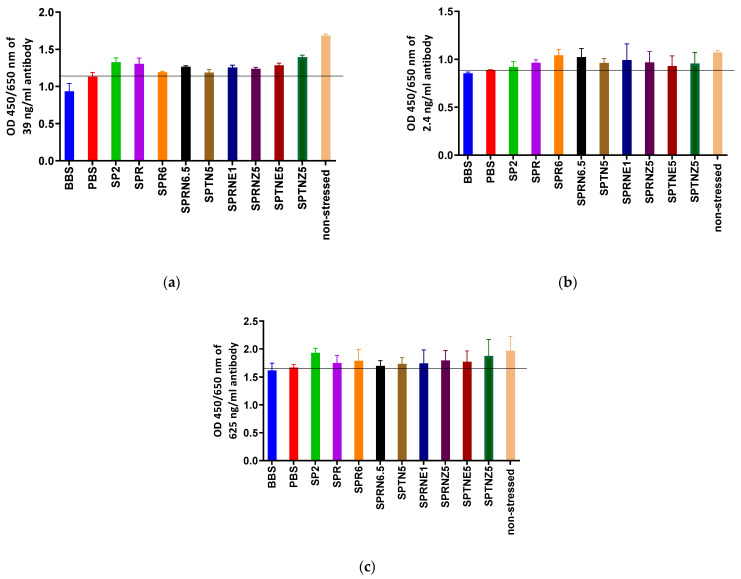
Two weeks 37 °C stability comparison of PRO-C3 (**a**), PRO-C6 (**b**), and CTX-III (**c**) antibodies stored in different buffers. The PBS groups were the benchmark, whereas the non-stressed groups were positive controls. CTX-III: C-terminal cross-linked telopeptides of type III collagen; PRO-C3: the C-terminus of procollagen III N-terminal propeptide; PRO-C6: the C-terminus of type VIa3 collagen. BBS: 0.1 M borate buffered saline, pH 8, 0.9% NaCl; PBS: 0.1 M PBS, pH 7.4, 0.9% NaCl; SP2: 0.9% NaCl; SPR: 37 mM citrate, 125 mM phosphate, pH 6.0; SPR6: 37 mM citrate, 125 mM phosphate, pH 6.0, 0.9% NaCl; SPRN6.5: 29 mM citrate, 142 mM phosphate, pH 6.5, 0.9% NaCl; SPTN5: 50 mM Na-citrate, pH 6.0, 0.9% NaCl; SPRNE1: 37 mM citrate, 125 mM phosphate, pH 6.0, 0.9% NaCl, 25% ethylene glycol; SPRNZ5: 37 mM citrate, 125 mM phosphate, pH 6.0, 0.9% NaCl, 0.05% sulfobetaine; SPTNE5: 50 mM Na-citrate, pH 6.0, 0.9% NaCl, 25% ethylene glycol; SPTNZ5: 50 mM Na-citrate, pH 6.0, 0.9% NaCl, 0.05% sulfobetaine; No-stressed: 0.1 M PBS, pH 7.4, 0.9% NaCl, stored at −20 °C until the test.

**Table 1 diagnostics-13-01835-t001:** Materials that impact lot-to-lot variance of immunoassays.

Materials	Specifications That May Lead to LTLV ^1^
** *Internal materials* **	
Antigen	Unclear and color appearance, low storage concentration, high aggregate, low purity, inappropriate storage buffer
Antibody	Unclear and color appearance, low storage concentration, high aggregate, low purity, inappropriate storage buffer
Enzyme	Inconsistent enzymatic activity
Conjugate	Unclear appearance, low concentration, low purity
Kit controls and calibrators	Kit controls use the same materials as the calibrators
Buffer/Diluent	Not mixed thoroughly, resulting in pH and conductivity deviation
Others (containers, microtiter plates, magnetic beads, etc.)	Unclean containers. Inhomogeneous magnetic beads
** *External materials* **	
Lot-to-Lot QC panel	Unstable and short shelf-life
Master calibrator	Not freeze-dried. Unstable

^1^ LTLV: lot-to-lot variance.

**Table 2 diagnostics-13-01835-t002:** Comparison of calibration curves using hybridoma and recombinant antibody against CTX-III ^1^.

Parameters	Hybridoma Antibody	Recombinant Antibody	Percent Deviation
Max signals (RLU ^2^)	493,180	412,901	−19.4%
Background (RLU)	4809	4546	−5.80%
EC50 ^3^ (ng/mL)	3.66	6.17	68.4%

^1^ CTX-III: C-terminal cross-linked telopeptides of type III collagen; ^2^ RLU: relative luminescence unit; ^3^ EC50: Half maximal effective concentration.

**Table 3 diagnostics-13-01835-t003:** Most popular additives and their proposed modes of action for minimizing protein aggregation. Adapted from Cytiva online course: Tips and troubleshooting: recombinant protein production.

Additive	Proposed Mode of Action
Glycerol, 5 to 40% (*v*/*v*) Sucrose, 10 to 40% (*w*/*v*) Glycine, 0.02 to 0.5 M Sorbitol, 5 to 40% (*w*/*v*)	Stabilizes native, intramolecular protein interactions
PEG ^1^, 1 to 15% (*v*/*v*) Nonionic detergents	Shields surface exposed hydrophobic sites (reduces protein-protein interactions)
Citrate, 0.02 to 0.4 M	Shields surface exposed hydrophobic sites (reduces protein-protein interactions)
Urea, up to 2 M Arginine, up to 2 M	Reduces protein-protein interactions
DTT ^2^, 0.1 to 1 mM	Prevents formation of intermolecular S-S bonds

^1^ PEG: Polyethene glycol. ^2^ DTT: Dithiothreitol.

## Data Availability

The data presented in this study are available on request from the corresponding author. The data are not publicly available due to restrictions on privacy.

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
