# Peer review of "Lot-to-Lot Variance in Immunoassays—Causes, Consequences, and Solutions"

_diagnostics, 2023, doi:10.3390/diagnostics13111835_

Round 1

Reviewer 1 Report

The article by Yunyun Luo and co-authors provides valuable information about decreasing between-lot variability of immunoassay kits. The most exciting thing about this paper is that researchers from IVD company share their experience on reproducibility issues and give recommendations on how to solve them.  As an academic researcher, I find this article catchy and insightful. The paper is well written and easy to read; it fits the scope of Diagnostics and will be of interest to readership. 

1. Line 34 - "in vitro" should be italicized

2. Lines 112-117 - text formatting

3. Lines 118-125 - please delete repeating fragments of Fig. 1. The same formatting issue with lines 343-353

4. In Table 2 differences in RLU between assays utilized hybridoma or recombinant antibodies are shown. What was difference between batches for hybriodoma-based assays and recombinant MAb-based assays?

5. Is it a common situation that hybridoma antibodies have better purity than recombinant counterparts?

Reviewer 2 Report

Major revision:

I don't think you have cited the necessary references for each part since this is a review article. Please cite references for any fact statement not from your study and send it back to me for further review.

Author Response

Actually, the manuscript is intended for submission as a viewpoint or tech report based on our 20-year immunoassay experience. However, the journal editor thinks the words exceed the maximum limits, so I was asked to change it to review.  

Reviewer 3 Report

The review is devoted to a detailed consideration of the causes of variations in the results of immunoassay due to the lot-to-lot variances. This is an actual problem faced by both researchers and those who use this method in clinical practice. The authors analyze in great detail different reasons that cause this variability, related to both the properties of biological components and the technical aspects of the production of kits. The great advantage of the review is a large section with valuable suggestions and approaches to overcoming possible LTLVs.

The review is well written and structured.

In my opinion, this review should be accepted for publication in the Diagnostics after correcting minor issues that are given below:

i)                   Line 55 - the comparison with the icing on the cake is unclear. After all, the quality of the cake is also important and does not depend on the quality of the icing. Perhaps you need to add a couple of explaining phrases.

ii)                 The formatting of the sections 2.1.3. and 3.2.1 should be checked.

iii)               The text and captions to Fig. 1 are mixed up.

iv)               Check the number of significant digits in Table 2 according to the deviation values. The EC50 units are not specified.

v)                 Check the reference 6. This is a review of Australian Commission, it is better to give its name without abbreviations.

Round 2

Reviewer 2 Report

Please do final English language and style check.